# Diagnosis in Bytes: Comparing the Diagnostic Accuracy of Google and ChatGPT 3.5 as an Educational Support Tool

**DOI:** 10.3390/ijerph21050580

**Published:** 2024-05-01

**Authors:** Guilherme R. Guimaraes, Ricardo G. Figueiredo, Caroline Santos Silva, Vanessa Arata, Jean Carlos Z. Contreras, Cristiano M. Gomes, Ricardo B. Tiraboschi, José Bessa Junior

**Affiliations:** 1Programa de Pós-Graduação em Saúde Coletiva, Universidade Estadual de Feira de Santana (UEFS), Feira de Santana 44.036-900, Brazil; guilhermeguimaraes304@gmail.com (G.R.G.); s.carolinne5@gmail.com (C.S.S.); vafigueiredo@uefs.br (V.A.); zambrano.jeancarlos@gmail.com (J.C.Z.C.); rbtiraboschi@uefs.br (R.B.T.); bessa@uefs.br (J.B.J.); 2Faculty of Medicine, Universidade de São Paulo (USP), São Paulo 01.246-904, Brazil; crismgomes@uol.com.br

**Keywords:** medical education, medical informatics applications, artificial intelligence, diagnosis, urology

## Abstract

Background: Adopting advanced digital technologies as diagnostic support tools in healthcare is an unquestionable trend accelerated by the COVID-19 pandemic. However, their accuracy in suggesting diagnoses remains controversial and needs to be explored. We aimed to evaluate and compare the diagnostic accuracy of two free accessible internet search tools: Google and ChatGPT 3.5. Methods: To assess the effectiveness of both medical platforms, we conducted evaluations using a sample of 60 clinical cases related to urological pathologies. We organized the urological cases into two distinct categories for our analysis: (i) prevalent conditions, which were compiled using the most common symptoms, as outlined by EAU and UpToDate guidelines, and (ii) unusual disorders, identified through case reports published in the ‘Urology Case Reports’ journal from 2022 to 2023. The outcomes were meticulously classified into three categories to determine the accuracy of each platform: “correct diagnosis”, “likely differential diagnosis”, and “incorrect diagnosis”. A group of experts evaluated the responses blindly and randomly. Results: For commonly encountered urological conditions, Google’s accuracy was 53.3%, with an additional 23.3% of its results falling within a plausible range of differential diagnoses, and the remaining outcomes were incorrect. ChatGPT 3.5 outperformed Google with an accuracy of 86.6%, provided a likely differential diagnosis in 13.3% of cases, and made no unsuitable diagnosis. In evaluating unusual disorders, Google failed to deliver any correct diagnoses but proposed a likely differential diagnosis in 20% of cases. ChatGPT 3.5 identified the proper diagnosis in 16.6% of rare cases and offered a reasonable differential diagnosis in half of the cases. Conclusion: ChatGPT 3.5 demonstrated higher diagnostic accuracy than Google in both contexts. The platform showed satisfactory accuracy when diagnosing common cases, yet its performance in identifying rare conditions remains limited.

## 1. Introduction

Since the introduction of early data processing machines in the 1940s, researchers across diverse disciplines have been captivated by their possible uses, with researchers in the medical field showing particular interest [1]. As early as 1959, Brodman and colleagues demonstrated that a trained computerized system could identify patterns in a group of symptoms reported by patients and suggest possible diagnoses, performing comparably to physicians receiving the same information [2]. From that point forward, data analytics technologies and Artificial Intelligence (AI) gained prominence in several medical fields, including public health, medical image analysis, and clinical trials, among others [3].

The increasing capability of these tools to integrate information has allowed researchers to envision diagnostic applications far beyond what Broadman presented. It is now possible to input a patient’s symptoms into everyday search tools and receive a list of likely diagnoses [3,4]. Tang and Ng explored this utility in 2006, when they assessed the frequency of accurate diagnoses provided when specific disease symptoms were searched on Google, the leading internet search site [5].

In addition to search tools, new AI chatbots have recently been used to explore this field [6]. Among the most notable advancements is OpenAI’s Generative Pre-trained Transformer, ChatGPT 3.5. Unlike conventional search engines that return pages based on keyword matching, ChatGPT 3.5 generates real-time responses, drawing from a vast database [7,8]. Internet users can access version 3.5 for free, as it is the most recent version available at no cost.

Given their capabilities for continuous and incremental learning, rapid summarization of textual data, and generation of natural language responses, large language models (LLMs) have been widely applied across various domains, notably in medical training [9]. These models efficiently assimilate and synthesize vast amounts of information, making them valuable tools for educational purposes in healthcare settings. In this setting, medical education has moved towards a competency-based education paradigm. Generative AI technologies have been increasingly employed in new competencies training for doctors and medical graduates [9]. AI-enhanced predictive models for risk stratification have shown great potential in the healthcare sector, especially in reducing diagnostic errors and improving patient safety.

In this context of evolving information technologies and their increasingly pervasive integration in medicine, this study aimed to assess the diagnostic accuracy and compare the agreement of two prominent online search tools, Google and ChatGPT 3.5, for urological conditions.

## 2. Materials and Methods

This pilot study was conducted between April and June 2023, in which the diagnostic properties of the Google and ChatGPT 3.5 platforms were assessed. These tools were evaluated based on the responses to 60 clinical cases related to urological pathologies, divided into “prevalent conditions” and “unusual conditions”. Questions were formulated in Portuguese.

In the first group, 30 descriptions summarized the typical clinical presentation of prevalent urological diseases, published by the European Association of Urology and UpToDate guidelines (Table 1). In the second group, the remaining 30 cases were based on reports published between 2022 and 2023 in Urology Case Reports, selected based on the typicality of their manifestations (Table 2). Questions requiring extensive evaluation for specific diagnoses were excluded.

Each clinical case was inputted into Google Search and ChatGPT 3.5, and the results were categorized as “correct diagnosis”, “likely differential diagnosis”, and “incorrect diagnosis”, according to the blind and random judgment of a panel of three experts. For Google Search, a new incognito window in the browser with no linked account was used to minimize any influence from previous search history, and the first three displayed results were considered for diagnostic categorization. For ChatGPT 3.5, a specific individual account was created to reduce the influence of prior searches. Regarding the use of AI or AI-assisted technologies, Google and ChatGPT 3.5 were employed exclusively to evaluate the diagnostic accuracy of each AI platform based on the responses to 60 clinical cases related to urological pathologies.

The findings of this study were described in absolute numbers and corresponding percentages. The Chi-square test was used for proportion comparison, and the Kappa test was employed to assess agreement between the instruments. “All tests are two-tailed, and *p* values < 0.05 were considered statistically significant”. We utilized GraphPad Prism version 10.0.0 for Windows, provided by GraphPad Software from Boston, MA, USA, for the statistical analysis and to create the graphical representations of the data.

## 3. Results

Both platforms showed promising results when dealing with prevalent urological cases in daily practice. Google Search demonstrated a correct diagnosis, likely differential diagnosis, and incorrect diagnosis in 16 (53.3%), 7 (23.3%), and 7 (23.3%) of the clinical cases. ChatGPT 3.5 outperformed Google, providing a correct diagnosis in 26 (86.6%) cases and offering likely differential diagnoses in 4 (13.3%) cases, with no incorrect diagnoses (*p* = 0.004).

Regarding the unusual urological conditions, the performances of the two platforms significantly diverged. Google could not provide any correct diagnosis but offered likely differential diagnoses in six (20%) cases. Google outputted an incorrect diagnosis in the remaining 24 (80%) cases. Nevertheless, ChatGPT 3.5 had moderate success in diagnosing rare conditions, with a correct diagnosis in five (16.6%) cases. Notably, it provided a likely differential diagnosis in 15 (50%) cases. However, it is important to note that the platform provided an incorrect diagnosis in 10 cases, constituting 33.3% of the instances evaluated (*p* = 0.0012). The results are summarized in Figure 1.

In a comparative accuracy analysis, ChatGPT 3.5 was significantly superior to Google. The platform provided correct diagnoses or proposed suitable differential diagnoses in 50 (83.3% 71.7–90.80 95%CI) out of 60 cases, in contrast to Google’s performance of 29 (48.3% 36.1–60.7 95%CI) out of 60 cases (OR = 3.62 1.50–8.73 95%CI *p* < 0.001). There was low agreement between both diagnostic instruments (Kappa = 0.315).

## 4. Discussion

This study showed that ChatGPT 3.5 demonstrated superior diagnostic capabilities compared to Google in real-life urological scenarios. Both platforms varied in performance depending on the complexity and rarity of urological conditions. While Google remained moderately effective in prevalent urological cases, its performance reduced significantly in unusual urological conditions. ChatGPT 3.5 showed high diagnostic accuracy in common urological diseases and moderate success in diagnosing rare and uncommon conditions.

In this comparative study, the LLMs demonstrated proficiency in extracting information and responding to structured inquiries, achieving accuracy rates ranging from 53.3% to 86.6%. ChatGPT, in particular, operates by sequentially predicting word fragments until a complete response is formed. Its architecture effectively processes and integrates complex clinical data, yielding contextually relevant interpretations [10]. In contrast, Google often retrieves more generalized information. ChatGPT utilizes a comprehensive, curated medical dataset, including peer-reviewed articles and clinical case studies, which enhances its ability to provide context-aware responses that adhere to contemporary medical standards. This specialized approach contributes to ChatGPT’s higher diagnostic accuracy in the nuanced field of urology. While ChatGPT is capable of incremental learning, allowing it to retain information from previous interactions to refine future responses [11], this feature was unlikely to influence the results in this study due to using a specific individual account designed to minimize the impact of prior searches.

The study outcomes corroborate the importance of acknowledging the fluctuating efficacy of these tools across various clinical contexts, highlighting both their advantages and the domains that require enhancement. These findings also raise essential questions about the role of AI-based platforms like ChatGPT 3.5 in clinical decision making and education. The exponential growth in medical knowledge exacerbates the challenges faced by healthcare professionals. Estimates suggest that the rate at which medical knowledge expands has significantly accelerated—from taking 50 years to double in 1950, down to just 73 days in 2020 [12]. These rapid advances mean medical students must master 342 potential diagnoses for 37 frequently presented signs and symptoms before graduating [13]. This amount of information can be overwhelming and underscores the growing need for accurate and more efficient diagnostic tools to assist physicians and other healthcare providers.

Internet search engines such as Google have been around for a while and have progressively been utilized within the healthcare and educational sectors [14,15]. In 1999, Graber and colleagues assessed the capacity of online search engines in resolving medical cases, finding that these platforms could correctly answer half of the questions [14]. By 2006, Google had already become the dominant online search tool, providing correct diagnoses in 58% of cases in Tang and Ng’s study [5]. Another study evaluated the diagnostic accuracy of medical students before and after consulting Google and PubMed, noting a not statistically significant but interesting 9.9% increase in diagnostic accuracy [16].

More recently, chatbots, which combine AI with messaging interfaces, have emerged as precise tools for generating direct responses [17,18]. A Japanese study showed that ChatGPT 3 achieved a correct diagnostic rate of 93.3% when considering a list of ten probable differential diagnoses [19]. The same platform demonstrated an accuracy rate of 80% in questions about cirrhosis and hepatocellular carcinoma and an accuracy of 88% in breast cancer prevention and screening [20,21].

In the field of urology, ChatGPT 3.5’s performance has been variable. In one study, the platform correctly answered 92% of pediatric urology questions [22]. However, the platform’s performance was reduced to 52% in another study on general urological cases [23]. As illustrated by Huynh and colleagues, ChatGPT 3.5 performed poorly in the American Urological Association’s 2022 self-assessment study program, scoring below 30% [24]. Studies evaluating the accuracy of information on urological conditions provided to the general public have shown that while the responses were generally acceptable, they also contained significant inaccuracies [25,26].

We are observing the beginning of a new era in the history of medical education. The COVID-19 pandemic rapidly increased the use of new technologies in medical competencies training. This surge in adoption coincides with the release of ChatGPT, which has gained widespread recognition in educational settings by both teachers and students. At the same time, there have been quick policy adjustments regarding AI’s role in writing and academic publishing [27]. Educational leaders now face the crucial task of understanding the extensive impact of these changes across all aspects of health education and ethical dilemmas [9].

The current study provides valuable insights into existing research by directly comparing the diagnostic effectiveness of Google and ChatGPT 3.5 for urological conditions. Our results indicate that ChatGPT 3.5 outperformed Google in common and rare cases. While the platform showed high precision in diagnosing common urological conditions, it demonstrated moderate success with rarer diseases. These findings offer a promising perspective for integrating such tools in medical education and clinical workflows.

These platforms can assist urological researchers in analyzing and interpreting data, improving grammar and clarity in scientific manuscripts, and creating educational material. The role of these tools should be viewed as supplemental to the expertise of doctors and healthcare professionals providing patient care. Given the identified limitations, it is evident that both platforms require improvements. The ongoing process of expanding access to medical databases and continuous algorithmic training will increase its future utility. Ethical considerations cannot be overlooked either. There must be extensive discussion about the quality of information these platforms provide and how user privacy is ensured, as sensitive data may be involved. As these platforms evolve, their utility as diagnostic tools may become more robust, promoting more innovative and secure healthcare applications.

Currently, LLMs face significant limitations in medical decision making and diagnostics, including a lack of access to copyrighted private databases, a propensity for generating inaccurate or fabricated information (“hallucinations”), the unpredictability of responses, and constraints on training datasets [10]. The choice of cases included in this study could influence outcomes if they are not representative of the typical spectrum of urological conditions seen in clinical practice. Both platforms’ performance can vary significantly based on how questions are phrased. A study’s reliability may be affected if the prompts do not accurately reflect typical user queries or clinical scenarios. Both ChatGPT and Google continually update their algorithms. The results might not reflect the performance of newer or updated versions of the models. Finally, the diagnosis categories were determined by a panel of experts whose judgments could introduce subjective bias. The criteria for categorization (correct diagnosis, likely differential diagnosis, incorrect diagnosis) may not be uniformly applied.

Despite these challenges, the future of LLMs in medical diagnostics looks promising due to rapid advancements in artificial intelligence and machine learning algorithms. These models are becoming more proficient at processing the extensive medical literature and patient data, continually improving their diagnostic algorithms. The inclusion of specialized datasets, such as those covering rare diseases or intricate clinical scenarios, enhances their accuracy and relevance in medical settings. As training becomes more comprehensive and detailed, LLMs’ capacity to deliver precise, contextually appropriate medical advice is expected to evolve, significantly enhancing clinical decision support and revolutionizing patient care outcomes. This research paves the way for further research on integrating AI-based tools such as ChatGPT into clinical practice. This approach aims for a quick, reliable diagnosis and potentially enhances patient outcomes while augmenting human expertise.

Our study contributes to a broader discussion about the evolving role of technology in healthcare, specifically in urology, where accurate and timely diagnosis is often critical to treatment success and patient well-being.

## 5. Conclusions

We demonstrated that ChatGPT 3.5 exhibited superior diagnostic accuracy compared to Google in prevalent and rare urological scenarios. ChatGPT 3.5 displayed acceptable accuracy in cases of habitual conditions but was still relatively limited in rare cases. Such findings allow us to glimpse some of the possible uses of these tools in educational and training processes. Access to medical databases and ongoing development can bring considerable advances, enabling even more robust, innovative, and secure tools and possibly assisting us in caring for people.

## Figures and Tables

**Figure 1 ijerph-21-00580-f001:**
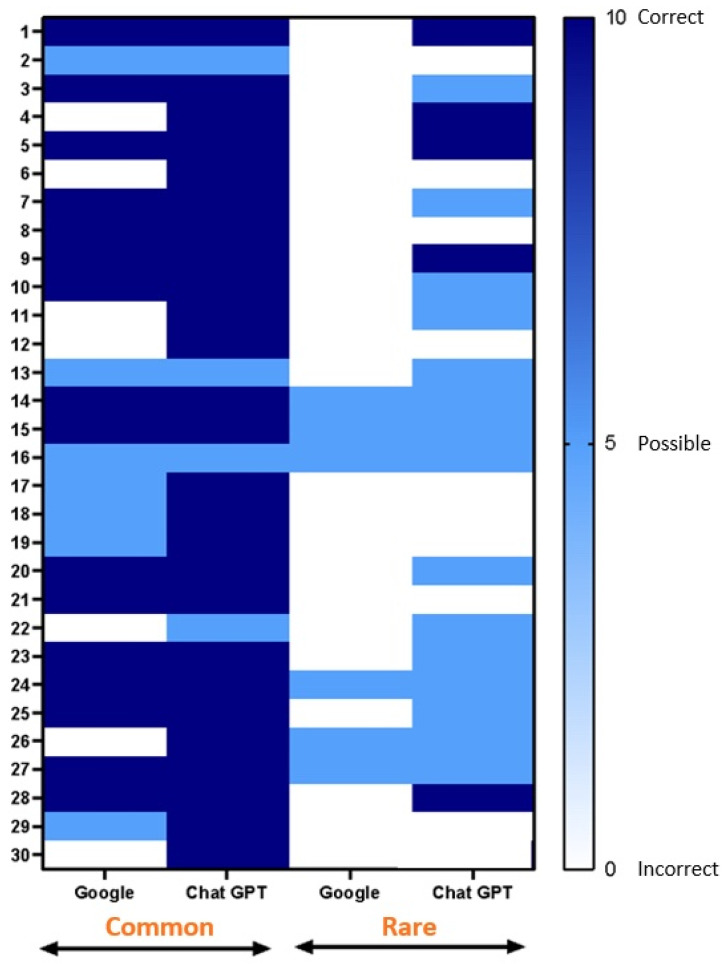
Diagnostic accuracy of Google and ChatGPT for urologic clinical cases. Common cases are represented on the left side, and rare cases are on the right. To determine the diagnostic accuracy of each platform, the outcomes were classified into three categories: correct diagnosis (dark blue), likely differential diagnosis (light blue), and incorrect diagnosis (white).

**Table 1 ijerph-21-00580-t001:** Prevalent urological conditions.

Pathology	Clinical Case
Case 1: UTI (Urinary Tract Infection)	Woman, 27 years old, complains of dysuria, polyuria, and hematuria for about 6 days. Reports episodes of unmeasured fever and right flank pain. What is the likely diagnosis?
Case 2: Nephrolithiasis (Kidney Stones)	35-year-old man with left flank pain for 3 days, colicky, of strong intensity radiating to the inguinal region on the same side, presents nausea, fever, and dysuria. What is the likely diagnosis?
Case 3: BPH (Benign Prostatic Hyperplasia)	45-year-old man presents dysuria associated with increased urinary frequency, urinary incontinence, and an enlarged prostate. What is the likely diagnosis?
Case 4: Prostate Cancer	62-year-old man reports urinary incontinence, blood in the urine, and discomfort when sitting for the past 1 and a half months. Lost 2 kg during this period. What is the likely diagnosis?
Case 5: Stress Urinary Incontinence	62-year-old man reports stress urinary incontinence, without complaints of urinary urgency. What is the likely diagnosis?
Case 6: Bladder Cancer	82-year-old man presents painless hematuria, associated with bladder irritability, increased urinary frequency, and urinary urgency. What is the diagnosis?
Case 7: Erectile Dysfunction	50-year-old man with frequent difficulty in achieving a satisfactory erection for sexual intercourse and difficulty in maintaining a satisfactory erection for penetration. What is the correct diagnoses?
Case 8: Varicocele	25-year-old man with a left scrotum resembling a bag of worms, scrotal pain, testicular atrophy, and difficulty in conceiving. What is the likely diagnosis?
Case 9: Interstitial Cystitis	45-year-old woman complains of pain and discomfort for 3 months in the suprapubic region, related to bladder filling, with relief upon urination. What is the correct diagnosis?
Case 10: Priapism	22-year-old sickle cell anemia patient complains of a rigid and painful erection for 5 h. What is the likely diagnosis?
Case 11: Kidney Cancer	63-year-old man reports moderate pain in the lower back, hematuria, sensation of incomplete emptying, and palpable abdominal mass. What is the diagnosis?
Case 12: Testicular Cancer	16-year-old boy complains of a solid lump palpated in the right testicle. He denies pain but reports a sensation of scrotal heaviness. What is the probable diagnosis?
Case 13: Urethral Stricture	45-year-old man complains of weak stream, polyuria, incomplete emptying, and post-micturition dribbling. What is the likely diagnosis?
Case 14: Phimosis	Pre-adolescent, 12 years old, comes to the outpatient clinic with his mother, complaining of excess preputial skin, with compromised retraction. He reports post-micturition balanitis, accompanied by erythema on the glans. What is the likely diagnosis?
Case 15: Hypospadias	Male newborn presents with dorsal hooded foreskin, abnormal penile curvature, and double urethral opening, one terminal and the other subcoronal. What is the likely diagnosis?
Case 16: Prostatitis	60-year-old man with high fever, dysuria, chills, irritating urinary symptoms, and cloudy urine. What is the likely diagnosis?
Case 17: Gonorrhea	23-year-old man, unmarried, sexually active with multiple partners, experiences burning during urination, greenish purulent discharge, and erythema. What is the possible diagnosis?
Case 18: Polycystic Kidney Disease	54-year-old newly diagnosed hypertensive man reports hematuria and compromised renal function. He mentions altered renal ultrasound and similar family history. What is the probable diagnosis?
Case 19: Nocturnal Enuresis (Bedwetting)	7-year-old boy presents with multiple episodes of nocturnal enuresis, started after his parents’ divorce 3 months ago. He denies other changes. What is the likely diagnosis?
Case 20: Urethritis	24-year-old man reports pain during urination, increased urinary frequency, urinary urgency, and transparent urethral discharge. What is the probable diagnosis?
Case 21: Male Infertility	26-year-old married man, trying to have children for over 18 months without success. Wife underwent tests with negative results for health problems. He underwent a semen analysis and obtained a count of 3 million sperm. What is the possible diagnosis?
Case 22: Urethral Diverticulum	34-year-old woman complains of urinary dribbling after urination, dysuria, and dyspareunia for 1 year, associated with palpable vaginal mass. She reports a history of pelvic surgery and denies other conditions. What is the probable diagnosis?
Case 23: Paraphimosis	23-year-old man presents with glandular edema and intense pain in the penis. Physical examination reveals a constricting band of tissue in the coronal sulcus. What is the probable diagnosis?
Case 24: Penile Cancer	60-year-old man presents with a palpable tumor in the glans of the penis, ulceration, and skin irritation for over six weeks. What is the probable diagnosis?
Case 25: Peyronie’s Disease	17-year-old man complains of severe dorsal penile curvature during erection, preventing penetration, associated with a palpable plaque on the dorsal side of the penis and pain. What is the probable diagnosis?
Case 26: Traumatic Kidney Injury	25-year-old man reports presence of blood in the urine, decreased urine volume, and mild abdominal pain after 24 h of hospitalization due to a motorcycle accident with lower rib fractures. What is the probable diagnosis?
Case 27: Cryptorchidism	4-month-old male child comes for a medical examination because the testicles were not noticed after birth and not at 2 months of age. What is the probable diagnosis?
Case 28: Female Sexual Dysfunction	29-year-old woman reports reduced sexual desire, anorgasmia, and dyspareunia for 5 years. She denies other associated conditions. What is the probable diagnosis?
Case 29: Neurogenic Bladder	70-year-old man complains of urinary incontinence, burning in the bladder region, and frequent urge to urinate in small amounts after a stroke. What is the probable diagnosis?
Case 30: Genital Herpes	19-year-old man reports the sudden appearance of multiple erythematous-based vesicles on the glans and dorsal side of the penis, painful and pruritic, for seven days. States progression to ulcers with scalloped borders. What is the probable diagnosis?

**Table 2 ijerph-21-00580-t002:** Unusual urological conditions.

Pathology	Clinical Case
Case 1: Melanoma in situ on the glans	28-year-old man complains of a brownish spot of about 8 mm on the penile glans, with an irregular central black point. He denies pain, and no alterations were found on physical examination. What is the likely diagnosis?
Case 2: Renal Papillary Hyperplasia	58-year-old man complains of urinating blood after exercise, denies coagulopathies, mentions NSAID use. What is the likely diagnosis?
Case 3: Urethral Stone	10-year-old child presents with acute urinary retention accompanied by penile pain, pelvic pressure, and a rigid, movable urethral mass.
Case 4: Penile Necrosis associated with Aortic Dissection	30-year-old man with lower limb paralysis, darkening of the scrotal sac and penis for 2 weeks, with loss of the urge to urinate and defecate. No hematuria or cloudy urine. Legs were blue.
Case 5: Renal Mucormycosis	56-year-old man presents with painful exophthalmos. Orbital tissue sampling revealed polymicrobial infection. Fungal polymerase test also revealed Rhizopus oryzae. A CT scan revealed an abscess in the lower right pole of the kidney. The abscess was drained, and the pus contained extended-spectrum beta-lactamase-producing enterobacteria. What is the likely diagnosis?
Case 6: Urachal Sinus (atypical case)	23-year-old woman is seen in the emergency room reporting lower abdominal pain and scant, foul-smelling umbilical discharge for 3 days. She does not report hematuria. Blood tests showed mild infection, and urinalysis and urine culture came back negative. What is the likely diagnosis?
Case 7: Ureterocoele Hernia with Gluteal Abscess	90-year-old woman with right buttock and hip pain, laboratory results showing signs of inflammation and mild renal dysfunction. Contrast-enhanced abdominal and pelvic CT scan reveals gluteal mass, hydronephrosis, and left-sided ureteral dilation with ureteral disconnection in the pelvis. Retrograde urography shows ureter folded in the left sciatic foramen. What is the likely diagnosis?
Case 8: Testicular Neoplasm	40-year-old man with symptoms of hyperandrogenism, absence of findings in the testicles on ultrasound and initial scrotal examination, and adrenal alteration. What is the possible diagnosis?
Case 9: Penile Fracture with Urethral Injury	37-year-old man arrives at the emergency room with complaints of pain, acute edema of the penis, rapid detumescence, blood discharge from the urethral meatus, and inability to urinate, with onset 3 h after sexual activity. Significant hematoma observed on penis during physical examination. Lab tests are normal. What is the likely diagnosis?
Case 10: Emphysematous Cystitis	78-year-old woman presents in the emergency room with signs of peritonitis associated with urinary incontinence and fever. Contrast-enhanced CT scan of the abdomen and pelvis reveals pneumoperitoneum, free fluid in the cavity, and air within the urinary bladder. What is the likely diagnosis?
Case 11: Urethral Diverticulum without Urethral Stricture	34-year-old man reports ejaculatory difficulty, urine loss after urination, and presents with penile-scrotal mass that increases during urination on physical examination. What is the likely correct diagnosis?
Case 12: Scrotoschisis	Male newborn, 2 days old, is brought to the emergency room with complete evisceration of the right testicle through a small defect in the right hemiscrotum wall. What is the likely diagnosis?
Case 13: Xanthogranulomatous Orchitis	77-year-old man presenting with right scrotal mass. He reports increasing scrotal swelling on the right side accompanied by pain for two days. Imaging exams revealed heterogeneous lesion involving the testicle, with collection in the underlying scrotal wall. What is the likely diagnosis?
Case 14: Laughing Urinary Incontinence	16-year-old female adolescent presenting with total and unstoppable urinary incontinence when laughing, normal menstrual pattern, and no other urinary complaints. Not responsive to previous anticholinergic therapy. What is the likely diagnosis?
Case 15: Wunderlich Syndrome	66-year-old man, without history of trauma and using antiplatelet medication, arrives at the emergency room with severe generalized abdominal pain and hemodynamic instability. Contrast-enhanced abdominal CT scan reveals perirenal and retroperitoneal hyperdensity, indicating hematoma. What is the likely diagnosis?
Case 16: Spermatocele Torsion	A 25-year-old man presented to the emergency room with sudden-onset right scrotal pain. On physical examination, he had a swollen and tender right hemiscrotum. Scrotal ultrasonography revealed testicular edema and a cystic mass originating from the right epididymis. What is the likely diagnosis?
Case 17: Testicular Tuberculosis	41-year-old man, pain and swelling in the right testicle, fever of 38° for 3 months, without ulcers. Painless, palpable, firm hypertrophied lymph nodes. Left scrotum normal, right scrotum painful with inflammatory signs, 2 cm swelling, painful, solid, not adhered to scrotal skin. What is the diagnosis?
Case 18: Duplicated Urethra	47-year-old man reports urinary incontinence, noticed double stream during urination, and frequent urinary infections. What is the possible diagnosis?
Case 19: Penile Sarcoidosis	A healthy man in his forties reports progressive inability to retract the foreskin over the past three weeks, associated with paresthesia in the glans and diffuse abdominal pain. On physical examination, a hardened area is noted on the proximal penile shaft, with multiple palpable nodules. He denies fever, dysuria, hematuria, and alterations in lab and imaging tests. What is the likely diagnosis?
Case 20: Proximal Ureter Rupture from Ureteral Catheterization	76-year-old woman with a history of right-sided abdominal pain, persistent fever, and vomiting for 1 day. He reports onset of symptoms after changing the routine of her long-term Foley catheter. He develops sepsis and acute kidney injury. What is the likely diagnosis?
Case 21: Urethral Diverticulum	23-year-old man with recurrent urinary infections, with dysuria associated with late dribbling, examination after micturition revealed floating penoscrotal pouch collapsing completely the urine output through the urethral meatus. What is the possible diagnosis?
Case 22: Penile Calciphylaxis	54-year-old man presents with penile and scrotal necrosis with a one-month evolution. He reports initial onset of penile pain and darkening of the glans, progressing to the penile shaft and scrotum. He reports a history of diabetes mellitus, hypertension, diabetic retinopathy, and chronic kidney disease on dialysis. What is the likely diagnosis?
Case 23: Hydronephrosis in Pelvic Kidney	18-year-old man with colicky lower abdominal pain and progressive inability to urinate, reaching anuria. He reports strong desire, but little elimination. What is the likely diagnosis?
Case 24: Acute Vasitis	Healthy 27-year-old man presents with left scrotal pain associated with inguinoscrotal swelling and nausea. Physical examination reveals edema extending along the left inguinal area. What is the likely diagnosis?
Case 25: Scrotal Basal Cell Carcinoma	58-year-old man presents with anterior scrotal lesion, 19 mm in its greatest diameter, for eight months. He reports that the lesion initially resembled a “pimple,” evolving with erosion and ulceration. He denies associated pruritus and reports a past medical history of condyloma acuminatum, smoking, intravenous drug use, stabilized psoriasis on biologic therapy, and two previously excised basal cell carcinomas on the back. What is the likely diagnosis?
Case 26: Castleman’s Disease (Lymphadenopathy) in the scrotum	79-year-old man reports the appearance of a painless mass in the right scrotum with significant growth in the past year. What is the likely diagnosis?
Case 27: Prolapsed Ectopic Ureterocele into the Vulva	12-month-old girl presents with red, smooth, prolapsed vulvar swelling through the urethral orifice. What is the likely diagnosis?
Case 28: Periurethral Abscess of the Corpus Spongiosum	42-year-old man presents with painful, floating, and tender mass in the proximal lower part of the penis, associated with dysuria, with a 20-day evolution. What is the likely diagnosis?
Case 29: Persistent Müllerian Duct Syndrome (PMDS)	38-year-old man was referred to the urology clinic due to left-sided hydronephrosis secondary to an abdominal mass. On physical examination, the patient had a hypoplastic empty scrotum, without palpable testicle and without surgical scars. Imaging exams revealed rudimentary uterus attached to a large mass that is replacing the left testicle. What is the likely diagnosis?
Case 30: Partial Thrombosis of the Corpus Cavernosum	25-year-old man reports pain and swelling in the penis and perineum, with penile sensitivity to touch especially on the left proximal side, and a seven-day evolution. He denies previous trauma. What is the likely diagnosis?

## Data Availability

Additional data will be available upon request.

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
