# Peer review of "Diagnosis in Bytes: Comparing the Diagnostic Accuracy of Google and ChatGPT 3.5 as an Educational Support Tool"

_ijerph, 2024, doi:10.3390/ijerph21050580_

Round 1
Reviewer 1 Report
Comments and Suggestions for Authors
Review Reports
A brief summary
This study aims to evaluate the diagnostic accuracy of Google and ChatGPT 3.5 for common and rare urological conditions. The result will contribute to the future application of AI-based tools into clinical practice in urology. The strength of this study is that it covers a variety of urological cases.
General concept comments
There are some points you should improve in the manuscripts (see specific comments). I think the biggest weakness of this manuscript is the discussion section. It does not have enough discussion associated with the result of this study. Why do you think the diagnostic accuracy of ChatGPT was superior to that of Google in urological conditions? What do you think the low agreement between Google and Chat GPT? Do you have any ideas or future perspectives based on the results?
Specific comments
Medical education has moved towards a competency-based education paradigm. Generative AI technologies have been increasingly employed in new competencies training for doctors and medical graduates. AI-enhanced predictive models for risk stratification have shown great potential in the healthcare sector, especially in reducing diagnostic errors and improving patient safety. “(Page 2, lines 53-57)
This paragraph does not seem to have clear relationship with the following paragraph. I recommend you delete this paragraph.
“This diagnostic accuracy study was…” (Page 2, line 65)
I do not think that the name of “diagnostic accuracy study” is common. I recommend you rephrase it (e.g., pilot study, experimental study).
“These 66 tools were evaluated based on the responses to 60 clinical cases related to urological pathologies… “(Page 2, lines 66-68)
How did you decide the sample size 60?
“Each clinical case was inputted into Google Search and ChatGPT 3.5...”(Page 2, line76)
Could you describe details about the output of ChatGPT, especially about the number of differential diagnoses ChatGPT made? Whether ChatGPT outputs single diagnosis or multiple differential diagnoses would largely influence the results.
“… “correct diagnosis”, “likely differential diagnosis”, and “incorrect diagnosis” …”(Page 2, lines 77-78)
Could you describe the definition of correct diagnosis, likely diagnosis, and incorrect diagnosis, respectively?
“… according to the blind and random judgment of a panel of three experts.” (Page 2, line 78)
Could you describe how you solved the inconsistency between three experts’ decisions?
“… the first three displayed results were considered for diagnostic categorization.” (Page 2, lines 80-81)
There might be some websites summarizing several diseases in a single place. How did you interpret those search results?
“The findings of the study were described in absolute numbers and corresponding percentages.” (Page 3, lines 86-87)
Could you show the 95% confidence interval?
“The Chi-square test was used for proportion comparison, …” (Page 3, line 87)
Could you show that the test was one-tailed or two tailed? In addition, you should mention about the P-value.
Result (Page 8, line 94)
In this section, you should describe the results in an objective manner (e.g., the rate of correct diagnosis, likely differential diagnosis, and incorrect diagnosis were XXX, XXX, and XXX, respectively). I recommend you avoid using subjective or ambiguous words (e.g., “over half of…”(line 97), “acceptable response”(line 99), “moderate success”(line 105), and “… did not offer a plausible diagnosis in the rest 108 of the cases” (lines 108-109) etc.). In addition, I recommend you summarize the results in a table.
“Both instruments failed to reach a high diagnostic accuracy in rare cases.” (Page 9, lines 113-114)
I think this sentence is unnecessary.
“In a comparative accuracy analysis…” (Page 9, line 110)
Could you show the results of the comparative accuracy analysis between Google and ChatGPT stratified by disease frequency (i.e., common vs rare)?
“Common cases are represented on the right side, and rare cases are on the left.” (Page 9. Line 119)
I thought that common cases are on the left side, and rare cases are on the right side on Figure 1.
Discussion (Page 9, line 123)
You should describe the limitation of this study at the end of this section.
“The study outcomes corroborate… …and ethical dilemmas” (Page 10, line132-Page11, line 171)
I recommend you move the contains of these paragraphs to the introduction section because these are some knowledge based on previous studies and not associated with the study results directly.
Comments on the Quality of English Language
English is appropriate and understandable.
Author Response
Response to Reviewers
We are grateful to the reviewer for their insightful comments on our paper. We have incorporated changes to reflect most of the reviewer's suggestions. Here is a point-by-point response to the reviewers' comments and concerns.
Reviewer 1
General concept comments
There are some points you should improve in the manuscripts (see specific comments). I think the biggest weakness of this manuscript is the discussion section. It does not have enough discussion associated with the result of this study. Why do you think the diagnostic accuracy of ChatGPT was superior to that of Google in urological conditions? What do you think the low agreement between Google and Chat GPT? Do you have any ideas or future perspectives based on the results?
Response: We agree with the reviewer that more quantitative information in the discussion session would add relevance for the reader. We have included two paragraphs to answer the reviewer’s questions, which we present below:
P.10 L.179:
In this comparative study, the large language models (LLMs) demonstrated proficiency in extracting information and responding to structured inquiries, achieving accuracy rates ranging from 53.3% to 86.6%. ChatGPT, in particular, operates by sequentially predicting word fragments until a complete response is formed. Its architecture effectively processes and integrates complex clinical data, yielding contextually relevant interpretations [Park, 2024]. In contrast, Google often retrieves more generalized information. ChatGPT utilizes a comprehensive, curated medical dataset, including peer-reviewed articles and clinical case studies, which enhances its ability to provide context-aware responses that adhere to contemporary medical standards. This specialized approach contributes to ChatGPT’s higher diagnostic accuracy in the nuanced field of urology. While ChatGPT is capable of incremental learning, allowing it to retain information from previous interactions to refine future responses [OpenAI. Introducing ChatGPT], this feature was unlikely to influence the results in this study due to using a specific individual account designed to minimize the impact of prior searches.
P.12 L.279:
Despite these challenges, the future of LLMs in medical diagnostics looks promising due to rapid advancements in artificial intelligence and machine learning algorithms. These models are becoming more proficient at processing extensive medical literature and patient data, continually improving their diagnostic algorithms. The inclusion of specialized datasets, such as those covering rare diseases or intricate clinical scenarios, enhances their accuracy and relevance in medical settings. As training becomes more comprehensive and detailed, LLMs' capacity to deliver precise, contextually appropriate medical advice is expected to evolve, significantly enhancing clinical decision support and revolutionizing patient care outcomes. This research paves the way for further research on integrating AI-based tools such as ChatGPT into clinical practice. This approach aims for a quick, reliable diagnosis and potentially enhances patient outcomes while augmenting human expertise.
OpenAI. Introducing ChatGPT [Internet]. [cited 2024 April 13]. Available from: https://openai.com/blog/chatgpt.
Park, YJ., Pillai, A., Deng, J. et al. Assessing the research landscape and clinical utility of large language models: a scoping review. BMC Med Inform Decis Mak 24, 72 (2024). https://doi.org/10.1186/s12911-024-02459-6
Specific comments
Medical education has moved towards a competency-based education paradigm. Generative AI technologies have been increasingly employed in new competencies training for doctors and medical graduates. AI-enhanced predictive models for risk stratification have shown great potential in the healthcare sector, especially in reducing diagnostic errors and improving patient safety. “(Page 2, lines 53-57)
This paragraph does not seem to have clear relationship with the following paragraph. I recommend you delete this paragraph.
Response: Although we agree with the reviewer that this paragraph was out of context, the discussion of medical training is a secondary objective of this study. We have revised the text, which we present below:
“Given their capabilities for continuous and incremental learning, rapid summarization of textual data, and generation of natural language responses, large language models (LLMs) have been widely applied across various domains, notably in medical training. These models efficiently assimilate and synthesize vast amounts of information, making them valuable tools for educational purposes in healthcare settings. In this setting, medical education has moved towards a competency-based education paradigm. Generative AI technologies have been increasingly employed in new competencies training for doctors and medical graduates9. AI-enhanced predictive models for risk stratification have shown great potential in the healthcare sector, especially in reducing diagnostic errors and improving patient safety.”
“This diagnostic accuracy study was…” (Page 2, line 65)
I do not think that the name of “diagnostic accuracy study” is common. I recommend you rephrase it (e.g., pilot study, experimental study).
Response: We revised the text according to the reviewer's comments using “pilot study” instead of “diagnostic accuracy study”
“These 66 tools were evaluated based on the responses to 60 clinical cases related to urological pathologies… “(Page 2, lines 66-68)
How did you decide the sample size 60?
Response: We explained case selection in the Methods section as follows: 30 descriptions summarized the typical clinical presentation of prevalent urological diseases, published by the European Association of Urology and UpToDate guidelines (Table 1). In the second group, the remaining 30 cases were based on reports published between 2022 and 2023 in Urology Case Reports, selected based on the typicality of their manifestations.
According to the European Association of Urology (EAU), these 30 specific cases represent typical clinical presentation of prevalent urological diseases. EAU incentives using these cases for medical training in the urology field. In a non-probabilistic approach, we have selected 30 case reports published between 2022 and 2023 in Urology Case Reports to match the first group as a sample of unusual urologic conditions.
“Each clinical case was inputted into Google Search and ChatGPT 3.5...”(Page 2, line76)
Could you describe details about the output of ChatGPT, especially about the number of differential diagnoses ChatGPT made? Whether ChatGPT outputs single diagnosis or multiple differential diagnoses would largely influence the results.
Response: We entered prompts into Google Search and ChatGPT 3.5 exactly as specified by the European Association of Urology and UpToDate guidelines, posing the question, “What is the likely/correct diagnosis?” The specific prompts used are listed in Tables 1 and 2. Each query generated a single primary diagnosis, barely with one or two likely differential diagnoses. The responses were classified into three categories: “correct diagnosis,” “likely differential diagnosis,” and “incorrect diagnosis,” based on a blind and random evaluation by a panel of three experts.
Could you describe how you solved the inconsistency between three experts’ decisions
The evaluation panel consists of three judges. Two primary judges independently assess each response. If these judges disagree on a response, the senior author, serving as the third judge, makes the final decision. We believe that this approach ensures rigorous evaluation and resolution of discrepancies while maintaining the integrity of the assessment process. The third judge, designated as the senior author, was not required to intervene in any of the assessments, as there were no disagreements between the primary judges throughout the evaluation process.
“… the first three displayed results were considered for diagnostic categorization.” (Page 2, lines 80-81)
There might be some websites summarizing several diseases in a single place. How did you interpret those search results?
Response: We are not unaware of any site that accurately covers a diverse range of urological conditions. We believe that both tools indeed search from all open-source published data.
“The findings of the study were described in absolute numbers and corresponding percentages.” (Page 3, lines 86-87)
Could you show the 95% confidence interval?
Response: In a comparative accuracy analysis, ChatGPT 3.5 was significantly superior to Google. The platform provided correct diagnoses or proposed suitable differential diagnoses in 50 (83.3% - 71.7 -90.80 95%CI) out of 60 cases, in contrast to Google's performance of 29 (48.3% - 36.1-60.7 95%CI) out of 60 cases (Diagnostic OR=5.31 p< 0.001). There was low agreement between both diagnostic instruments (Kappa=0.315).
“The Chi-square test was used for proportion comparison, …” (Page 3, line 87).
Could you show that the test was one-tailed or two tailed? In addition, you should mention about the P-value.
Response: We used chi-square to compare these two proportions. Chi-square tests are two-tailed, and p values <0.05 were considered statistically significant.
Result (Page 8, line 94) In this section, you should describe the results in an objective manner (e.g., the rate of correct diagnosis, likely differential diagnosis, and incorrect diagnosis were XXX, XXX, and XXX, respectively). I recommend you avoid using subjective or ambiguous words (e.g., “over half of…”(line 97), “acceptable response”(line 99), “moderate success”(line 105), and “… did not offer a plausible diagnosis in the rest 108 of the cases” (lines 108-109) etc.). In addition, I recommend you summarize the results in a table.
Response: We have revised the text, which we present below:
Both platforms showed promising results when dealing with prevalent urological cases in daily practice. Google Search demonstrated a correct diagnosis, likely differential diagnosis, and incorrect diagnosis in 16 (53,3%), 7 (23,3%), and 7 (23,3%) of the clinical cases. ChatGPT 3.5 outperformed Google, providing a correct diagnosis in 26 (86.6%) cases and offering likely differential diagnoses in 4 (13.3%) cases, with no incorrect diagnoses. Regarding the unusual urological conditions, the performances of the two platforms significantly diverged. Google could not provide any correct diagnosis but offered likely differential diagnoses in six (20%) cases. Google outputted an incorrect diagnosis in the remaining 24 (80%) cases. Nevertheless, ChatGPT 3.5 had moderate success in diagnosing rare conditions, with a correct diagnosis in 5 (16.6%) cases. Notably, it provided a likely differential diagnosis in 15 (50%). However, it is important to note that the platform provided an incorrect diagnosis in 10 cases, constituting 33.3% of the instances evaluated (Figure 1).
Figure 1 already summarizes the results. We cannot add another table since we have reached the editorial limit of tables and figures in the manuscript.
“Both instruments failed to reach a high diagnostic accuracy in rare cases.” (Page 9, lines 113-114)
I think this sentence is unnecessary.
Response: Revised accordingly
“In a comparative accuracy analysis…” (Page 9, line 110)
Could you show the results of the comparative accuracy analysis between Google and ChatGPT stratified by disease frequency (i.e., common vs rare)?
Response: We added the value of p after the descriptions of the hit rates for common and rare cases."
“Common cases are represented on the right side, and rare cases are on the left.” (Page 9. Line 119)
I thought that common cases are on the left side, and rare cases are on the right side on Figure 1.
Response: Revised accordingly
Discussion (Page 9, line 123)
You should describe the limitation of this study at the end of this section.
“The study outcomes corroborate… …and ethical dilemmas” (Page 10, line132-Page11, line 171)
I recommend you move the contains of these paragraphs to the introduction section because these are some knowledge based on previous studies and not associated with the study results directly.
Response: We agree with the reviewer that information about a study’s limitations would add relevance for the reader. We have included a paragraph in the discussion session, which we present below:
P.11 L.254:
“Currently, LLMs face significant limitations in medical decision-making and diagnostics, including a lack of access to copyrighted private databases, a propensity for generating inaccurate or fabricated information ("hallucinations"), the unpredictability of responses, and constraints on training datasets10. The choice of cases included in the study could influence outcomes if they are not representative of the typical spectrum of urological conditions seen in clinical practice. Both platforms' performance can vary significantly based on how questions are phrased. The study's reliability may be affected if the prompts do not accurately reflect typical user queries or clinical scenarios. Both ChatGPT and Google continually update their algorithms. Results might not reflect the performance of newer or updated versions of the models. Finally, The diagnosis categories were determined by a panel of experts whose judgments could introduce subjective bias. The criteria for categorization (correct diagnosis, likely differential diagnosis, incorrect diagnosis) may not be uniformly applied.”
“The study outcomes corroborate… …and ethical dilemmas” (Page 10, line132-Page11, line 171)
I recommend you move the contains of these paragraphs to the introduction section because these are some knowledge based on previous studies and not associated with the study results directly.
Response: Although we appreciate the reviewer's perspective, we believe that ethical considerations are a fundamental concern in the implementation of large language models (LLMs) in medical diagnostics. These issues are intrinsically linked to the widespread application of our study’s findings in the diagnostic and medical training fields. Therefore, we respectfully disagree with the suggestion to relocate this discussion to the introduction. Instead, we propose keeping it in the discussion section. We implemented minor necessary adaptations to ensure that it aligns more closely with the central context of our findings, thereby emphasizing its importance and relevance to the overarching implications of our research.
“The study outcomes corroborate the importance of acknowledging the fluctuating efficacy of these tools across various clinical contexts, highlighting both their advantages and the domains that require enhancement. These findings also raise essential questions about the role of AI-based platforms like ChatGPT 3.5 in clinical decision-making and education. The exponential growth of medical knowledge exacerbates the challenges faced by healthcare professionals. Estimates suggest that the rate at which medical knowledge expands has significantly accelerated—from taking 50 years to double in 1950, down to just 73 days in 202012. These rapid advances mean medical students must master 342 potential diagnoses for 37 frequently presented signs and symptoms before graduating13. This amount of information can be overwhelming and underscores the growing need for accurate and more efficient diagnostic tools to assist physicians and other healthcare providers.”

Reviewer 2 Report
Comments and Suggestions for Authors
Hi, thank you for submitting this paper. It covers a topic that I had been thinking about, and it was very timely to see your study, particularly the part about performance in common and uncommon diagnosis. It was a very quick and to the point paper, and addresses a questions that is not addressed much in the literature that I have seen. I have one moderate suggestion to consider. The paper uses ChatGPT3.5. In the time interval since your study, numerous other LLMs/Chatbots have come out, such as ChatGPT4.0 and others, and these newer models generally outperform ChatGPT3.5 by a significant margin. I think the results of your study are still valid, that even ChatGPT3.5 outperforms google search, but I would have liked see some language to acknowledge the fact that the current LLM/Chatbot ecosystem is very different than when the study was done - but that the results are still relevant.
Comments on the Quality of English LanguageOne minor potential error. In the first paragraph of the results suction, you wrote: "However, it could have provided an acceptable response in the remaining cases". Did you mean "However, it could not provide an acceptable response...."?
Author Response
Response to Reviewers
We are grateful to the reviewer for their insightful comments on our paper. We have incorporated changes to reflect most of the reviewer's suggestions. Here is a point-by-point response to the reviewers' comments and concerns.
Reviewer 2
Comments and Suggestions for Authors
Hi, thank you for submitting this paper. It covers a topic that I had been thinking about, and it was very timely to see your study, particularly the part about performance in common and uncommon diagnosis. It was a very quick and to the point paper, and addresses a questions that is not addressed much in the literature that I have seen. I have one moderate suggestion to consider. The paper uses ChatGPT3.5. In the time interval since your study, numerous other LLMs/Chatbots have come out, such as ChatGPT4.0 and others, and these newer models generally outperform ChatGPT3.5 by a significant margin. I think the results of your study are still valid, that even ChatGPT3.5 outperforms google search, but I would have liked see some language to acknowledge the fact that the current LLM/Chatbot ecosystem is very different than when the study was done - but that the results are still relevant.
Response: Although we appreciate the reviewer's perspective, this study's design is most appropriate for addressing our research question. Since this study was performed in a low-middle-income country, we compared the two most popular free-of-charge LLMs for medical diagnosis. We included ChatGPT version 3.5, which remains the most advanced version available without cost, whereas ChatGPT 4.0 requires a fee. Although recent findings indicate superior diagnostic accuracy in paid versions of LLMs, the implications of our study are more relevant for communities with limited financial resources, such as those in Latin America. This focus ensures that our results are applicable and beneficial to regions where cost considerations significantly influence technological access and healthcare decisions.
The current study provides valuable insights into existing research by directly comparing the diagnostic effectiveness of Google and ChatGPT 3.5 for urological conditions.
Comments on the Quality of English Language
One minor potential error. In the first paragraph of the results suction, you wrote: "However, it could have provided an acceptable response in the remaining cases". Did you mean "However, it could not provide an acceptable response...."?
Response: We agree with the reviewer that this sentence needed more clarity. We have revised the text, which we present below:
Both platforms showed promising results when dealing with prevalent urological cases in daily practice. Google Search demonstrated a correct diagnosis, likely differential diagnosis, and incorrect diagnosis in 16 (53,3%), 7 (23,3%), and 7 (23,3%) of the clinical cases. ChatGPT 3.5 outperformed Google, providing a correct diagnosis in 26 (86.6%) cases and offering likely differential diagnoses in 4 (13.3%) cases, with no incorrect diagnoses. Regarding the unusual urological conditions, the performances of the two platforms significantly diverged. Google could not provide any correct diagnosis but offered likely differential diagnoses in six (20%) cases. Google outputted an incorrect diagnosis in the remaining 24 (80%) cases. Nevertheless, ChatGPT 3.5 had moderate success in diagnosing rare conditions, with a correct diagnosis in 5 (16.6%) cases. Notably, it provided a likely differential diagnosis in 15 (50%). However, it is important to note that the platform provided an incorrect diagnosis in 10 cases, constituting 33.3% of the instances evaluated (Figure 1).

Reviewer 3 Report
Comments and Suggestions for Authors
Please find my reviews attached.

Author Response
Response to Reviewers
We are grateful to the reviewer for their insightful comments on our paper. We have incorporated changes to reflect most of the reviewer's suggestions. Here is a point-by-point response to the reviewers' comments and concerns.
Reviewer 3
Review of "Diagnosis in Bytes: Comparing the Diagnostic Accuracy of Google and ChatGPT 3.5 as an Educational Support Tool " Date: April 3, 2024 The study aimed to assess the diagnostic precision of two widely used internet search tools: Google and ChatGPT 3.5, focusing on two categories of urological cases: symptoms-based and unusual cases. Utilizing gold standard diagnoses provided by medical experts, the authors aimed to compare diagnostic accuracy across platforms. However, in my opinion, the research appears to address a well-established, known hypothesis. The authors cited previous studies examining Google search’s diagnostic capabilities. However, these studies are outdated, spanning back 14-15 years. In the current landscape, with the emergence of further advanced language-based AI systems (such as ChatGPT 4, BARD, BING AI, etc.) researchers have shifted focus from comparing with Google to assessing these newer platforms 1-4 . Consequently, this study may lack contemporary relevance in the evaluation of diagnostic accuracy.
Response: Although we appreciate the reviewer's perspective, this study's design is most appropriate for addressing our research question. Since this study was performed in a low-middle-income country, we compared the two most popular free-of-charge LLMs for medical diagnosis. We included ChatGPT version 3.5, which remains the most advanced version available without cost, whereas ChatGPT 4.0 requires a fee. Although recent findings indicate superior diagnostic accuracy in paid versions of LLMs, the implications of our study are more relevant for communities with limited financial resources, such as those in Latin America. This focus ensures that our results are applicable and beneficial to regions where cost considerations significantly influence technological access and healthcare decisions.
The current study provides valuable insights into existing research by directly comparing the diagnostic effectiveness of Google and ChatGPT 3.5 for urological conditions.

Round 2
Reviewer 1 Report
Comments and Suggestions for Authors
Review Reports
A brief summary
This study aims to evaluate the diagnostic accuracy of Google and ChatGPT 3.5 for common and rare urological conditions. The result will contribute to the future application of AI-based tools into clinical practice in urology. The strength of this study is that it covers a variety of urological cases.
General concept comments
N/A
Specific comments
“Diagnostic OR=5.31 p< 0.001” (Page 9, line 123)
Could you add the 95% confidence interval?
Author Response
Response to Reviewer
We are grateful to the reviewer for their insightful comments on our paper. We have incorporated changes to reflect most of the reviewer's suggestions. Here is a point-by-point response to the reviewers' comments and concerns.
Specific comments
Page 9, line 123
Could you add the 95% confidence interval?
Response: Revised Accordingly
"In a comparative accuracy analysis, ChatGPT 3.5 was significantly superior to Google. The platform provided correct diagnoses or proposed suitable differential diagnoses in 50 (83.3% 71.7-90.80 95%CI) out of 60 cases, in contrast to Google's performance of 29 (48.3% 36.1-60.7 95%CI) out of 60 cases (OR=3.62 1.50-8.73 95%CI p< 0.001). There was low agreement between both diagnostic instruments (Kappa=0.315)."

Reviewer 3 Report
Comments and Suggestions for Authors
No further comments
Author Response
Response to Reviewer
No further comments
Response: We are grateful to the reviewer for their insightful comments on our paper. We have incorporated changes to reflect the reviewer's suggestions in the attached manuscript.
Best regards